# Non-Nutritive Sweeteners and Their Implications on the Development of Metabolic Syndrome

**DOI:** 10.3390/nu11030644

**Published:** 2019-03-16

**Authors:** Iryna Liauchonak, Bessi Qorri, Fady Dawoud, Yatin Riat, Myron R. Szewczuk

**Affiliations:** 1Graduate Diploma and Professional Master in Medical Sciences, Postgraduate Medical Education, School of Medicine, Queen’s University, Kingston, ON K7L 3N6, Canada; iryna.liauchonak@gmail.com (I.L.); Fady.Dawoud@hotmail.com (F.D.); riatyatin@gmail.com (Y.R.); 2Department of Biomedical and Molecular Sciences, Queen’s University, Kingston, ON K7L 3N6, Canada; bessi.qorri@queensu.ca

**Keywords:** non-nutritive sweeteners, type 2 diabetes mellitus, gut microbiota, GPCR, insulin receptor signaling, miRNAs

## Abstract

Individuals widely use non-nutritive sweeteners (NNS) in attempts to lower their overall daily caloric intake, lose weight, and sustain a healthy diet. There are insufficient scientific data that support the safety of consuming NNS. However, recent studies have suggested that NNS consumption can induce gut microbiota dysbiosis and promote glucose intolerance in healthy individuals that may result in the development of type 2 diabetes mellitus (T2DM). This sequence of events may result in changes in the gut microbiota composition through microRNA (miRNA)-mediated changes. The mechanism(s) by which miRNAs alter gene expression of different bacterial species provides a link between the consumption of NNS and the development of metabolic changes. Another potential mechanism that connects NNS to metabolic changes is the molecular crosstalk between the insulin receptor (IR) and G protein-coupled receptors (GPCRs). Here, we aim to highlight the role of NNS in obesity and discuss IR-GPCR crosstalk and miRNA-mediated changes, in the manipulation of the gut microbiota composition and T2DM pathogenesis.

## 1. Introduction

Artificial sweeteners have gained increasing attention as dietary assessment tools to help combat the obesity epidemic by providing a sweet taste without the extra calories [1]. Taste has a significant role in human perception of food quality, contributing to its overall pleasure and enjoyment. To this end, the development of sweeteners as food additives that mimic the sweet taste of natural sugars suggest promise [2]. These artificial sweeteners are classified as nutritive or nonnutritive, both of which enhance the flavor and texture of food. Nutritive sweeteners contain carbohydrates and provide calories (energy). Non-nutritive sweeteners (NNS) are very low calorie or zero calorie alternatives that provide minimal or no carbohydrates or energy [3].

As part of dietary intake, NNS consumption can modulate energy balance, and metabolic functions through several peripheral and central mechanisms, suggesting that NNS are not inert compounds as once thought [4]. However, the specific mechanism(s) and details of the effects of NNS consumption on host metabolism and energy homeostasis remain to be elucidated. This is particularly relevant as NNS have been an option for individuals to improve their health; yet, NNS consumption has been associated with increased risk factors for metabolic syndrome [5]. Here, metabolic syndrome refers to the collection of physiological, biochemical, clinical, and metabolic factors that contribute to the increased risk of cardiovascular disease and type 2 diabetes melitus (T2DM) [6]. Based on measurements and laboratory tests, metabolic syndrome can also contribute to hypertension, glucose intolerance, proinflammatory state, atherogenic dyslipidemia, prothrombic state [7], and kidney disease [8]. It is noteworthy that the cause of these health-related issues may be due to emerging contaminants in the environment worldwide and their associated risks to human health and the environment [9]. Interestingly, one study identified a total of 24 non-nutritive artificial sweeteners studies to their occurrence in the environment from 38 locations globally across Europe, including the United Kingdom, Canada, United States, and Asia. Overall, the findings of the study indicated that non-nutritive artificial sweeteners are present in surface water, tap water, groundwater, seawater, lakes, and atmosphere [9]. Furthermore, in a Norwegian pregnancy cohort study, sucrose-sweetened soft beverages were reported to increase the risk of congenital heart defects (CHDs) in offspring, while fruit juices, cordial beverages, and artificial sweeteners had no associations with CHD [10].

In this review, we discuss the current status on the use of non-nutritive sweeteners, or non-caloric artificial sweeteners (NAS) which are used interchangeably and the future use of NNS in the food industry. This review has a particular focus on identifying the underlying mechanisms that can be responsible for the development of metabolic syndrome associated with NNS consumption. The physiological effects of NNS including NNS-induced metabolic changes will be discussed. We will highlight the development of metabolic syndrome that collectively involves the potential role of GPCR-IR crosstalk in the development of glucose intolerance and insulin resistance, the development of T2DM and the pathological mechanisms by which microRNAs (miRNAs) may mediate the changes in gut microbiota composition.

## 2. Current Status on the Use of Non-Nutritive Sweeteners

Currently, the Food and Drug Administration (FDA) has approved the use of acesulfame-potassium (Ace-K), aspartame, neotame, saccharin, sucralose, and stevia (https://www.fda.gov/food/ingredientspackaginglabeling/foodadditivesingredients/ucm397725.htm). Saccharin was discovered as early as 1876 and was the “original” artificial sweetener used in the food industry. Unfortunately, saccharin and many of its sweet alternatives have been considered to be health hazards, and as a result, are banned in many countries. Recently, other sweeteners have been developed and implemented within the food industry. In general, there are three primary types of sweeteners used in the food industry today: high-intensity sweeteners (e.g., acesulfame potassium, advantame, aspartame, neotame, saccharin, and sucralose), sugar alcohols (e.g., erythritol, glycerol, mannitol, sorbitol, and xylitol), and natural sweeteners (e.g., honey, lucuma powder, maple syrup, monk fruit known as *Siraitia grosvenorii* swingle fruit extract, stevia, and yacon syrup) [11]. These sweeteners and their uses in the food industry are summarized in Table 1. The high-intensity sweeteners can be synthetic or natural and are classified into two categories: nutritive and non-nutritive. The majority of high-intensity sweeteners used today fall into the non-nutritive category, with the exception of aspartame. Sugar alcohols are found naturally in small amounts in fruits and vegetables but are produced commercially in larger quantities.

Although sugar substitutes have been around since the 1880s, artificial sweetener consumption has dramatically increased over the last two decades as they are favorable alternatives to sucrose and other sugar substitutes. NNS can be several hundred to thousands times sweeter than sucrose with negligible caloric value, making them favorable health tools in attempts to control caloric intake and to assist in weight loss [12,13]. This trend has resulted in NNS becoming a staple in the Western diet, with cross-sectional studies reporting that 25% of children and 41% of adults consume low-calorie sweeteners. Consumption of NAS is found to be higher amongst females, obese individuals, and non-Hispanic white individuals as well as those with higher incomes [12,14].

Although these low-calorie sugar substitutes seem promising, NAS consumption has been associated with several inconsistent reports regarding their effects on the body. Due to the up-and-down history surrounding sweeteners used in the food industry, it can be quite confusing to understand what they are and how they are used. The greatest concerns are regarding the safety and side effects associated with NAS consumption [15]. For example, artificial sweeteners were once thought to be good options for diabetic or obese individuals where they were safe to use, providing sweetness without added calories [3,16]. However, most sweeteners have been shown to have no beneficial effects on diabetes mellitus, with the possibility of increasing risk of the disease diabetes. There are also some concerns with regard to the increased risk of developing cancer [16] and kidney disease [8]. NAS safety and health benefits remain to be a topic of controversy due to the increased incidence of obesity and T2DM that parallel increased consumption of artificial sweeteners over the past decade [14,17]. Using the rapid evidence mapping (rEM) approach, Lam et al. identified a lack of studies assessing appetite and dietary intake-related outcomes in people with diabetes [18]. This approach required approximately 100 person-hours conducted over seven calendar months. It is thought that non-nutritive sweeteners provide fewer calories per gram than sucrose as they are not entirely absorbed by the digestive system [19].

## 3. Future of Artificial Sweeteners in the Food Industry

There are now growing concerns over obesity and other health issues, and as a result, there will be a demand for sweet alternatives. Consumers can be classified broadly into two categories: Those that are interested in having low-sugar, low-calorie options to promote a healthy lifestyle and to avoid some of the health issues associated with consuming high amounts of sugar, such as obesity, diabetes, and heart disease.Those who already have with one or more of these health issues and are looking for ways to improve their diet and manage their health.

While the demand for artificial sweetener options in the beverage industry has been high, the demand for low-calorie sweeteners in place of sugar in baked goods, candies, and ice cream is increasing [20]. This high consumer pool opens a larger market for food manufacturers, making it increasingly important to understand artificial sweeteners and the roles they play in the lives of consumers worldwide. The preferences for specific sweeteners may impact food and beverage sales, so it is important that manufacturers stay abreast of the scientific developments surrounding each sweetener and what their impact may have on the demand for that specific sweetener.

Despite FDA approval of several sucrose alternatives marked as Generally Recognized As Safe (GRAS), there remains growing concern about the potentially harmful side effects associated with NNS consumption. Although several epidemiologic studies are focusing on artificial sweetener use and weight gain, it is critical that when interpreting such studies we consider factors that affect causality, and control for confounding factors such as age, diet, and environment, as well as additional stressors that may modify microbiota composition [21]. The gaps in our knowledge regarding how NNS consumption is implicated in host metabolism reinforces the importance of research needed to understand the mechanistic action of NNS on the body.

## 4. Physiological Effects of Non-Nutritive Sweeteners

Increased incidence of obesity and diabetes make NNS and their low caloric value even more favorable diet supplements. It is generally accepted that high sugar diets contribute to metabolic disorders [22]. The National Heart, Lung and Blood Institute (NHLBI) define metabolic syndrome as the group of risk factors that would increase heart disease and other health problems such as diabetes and stroke. The metabolic risk factors include abdominal obesity, high triglyceride level, low HDL cholesterol level, high blood pressure, and high fasting blood sugar. High sugar diets have been associated with the development of insulin resistance, T2DM, and additional cardiovascular diseases that fall within the realm of metabolic syndrome [23]. Briefly, these conditions are a result of dietary sugar upregulating hepatic uptake and metabolism of fructose, which leads to liver lipid accumulation, dyslipidemia, decreased insulin sensitivity, and increased uric acid levels [24]. The role of non-nutritive sweeteners in metabolic syndrome has been discussed in other reviews, with a focus on three potential mechanisms: NNS interacting with sweet taste receptors, NNS interfering with gut microbiota composition, and NNS interfering with learned responses to sweetness [25]. These three mechanisms are depicted in Figure 1 and will be the focus of the remainder of this review.

## 5. Non-Nutritive Sweeteners Interact with Sweet-Taste Receptors

### 5.1. Sweet-Taste Receptors in the Mouth: Perception of Sweetness

The innate universal preference for sweetness once served to support survival as it was associated with food reward and energy (calories) in the form of carbohydrates; however, sweetness is now often delivered via added sugars [26]. Sweet taste perception first begins at the level of type 2 taste receptor cells (TRCs) clustered in taste buds on the tongue that are G protein-coupled receptors (GPCRs) [27]. There are two classes of GPCRs that have been identified: the taste 1 receptor (T1R) and taste 2 receptor (T2R) families [28]. Within the T1R family, the T1R2 and T1R3 subtypes have been found to form heterodimers that act as sweet-taste receptors [29].

Interestingly, the T1R2/T1R3 receptors recognize all of the chemically diverse compounds that are perceived as sweet by humans, including nutritive and non-nutritive sweeteners [29]. Given the vast number of compounds that can bind to the sweet-taste receptors, it is not surprising that there are different functional roles of T1R2 and T1R3 with multiple ligand binding sites corresponding to the many possible ligands [30,31]. Sweet-taste receptor signaling has been extensively studied and reported [32,33,34]. Since sweet-taste receptors are GPCRs, they can induce the downstream activation of second messenger systems that ultimately result in increased intracellular calcium levels and neurotransmitter release [31,35]. Briefly, when a sweet-tasting compound binds to the T1R2/T1R3 receptors, α-gustducin is activated. The GPCR Gα-gustducin was previously identified as the first protein molecularly associated with taste cells [36], but its role in taste signal transduction is still not completely understood. Gustducin has considerable sequence homology to transducin, which is also expressed in taste buds [37,38]. Both α-gustducin and α-transducin are known to activate a phosphodiesterase (PDE) and decrease intracellular cAMP levels. There is also an increase in phospholipase Cβ2 (PLCβ2) concentration which in turn increases production of inositol 1,4,5-trisphosphate and diacylglycerol. These compounds, in turn, activate the transient receptor potential cation channel subfamily M member 5 (TRPM5), which subsequently increase intracellular calcium and neurotransmitter release [32,39].

### 5.2. Sweet-Taste Receptors in the Gut: Effect of Sweeteners on Hormone Secretion

Sweet-taste receptors have also been found throughout the gastrointestinal (GI) tract, the biliary tract, and the respiratory tract, suggesting that non-nutritive sweeteners have additional effects in the body and may not be the inert compounds that they were once thought to be [31,40,41,42]. Within the GI tract, sweet-taste receptors were primarily found in enteroendocrine L and K cells which secrete specific hormones, as well as in pancreatic β-islet cells [31,39]. These studies have shown that ligand binding to sweet taste receptors on enteroendocrine cells (EECs) in part affects hormone secretion. In particular, the use of a sweet-taste inhibitor decreased glucagon-like peptide-1 (GLP-1) and peptide YY (PYY) secretion by L cells, without affecting cholecystokinin (CCK) secretion from I cells, which are known to not express sweet-taste receptors [41,43]. Thus, it appears that this network of sweet taste signaling pathways in the oral cavity and the GI tract mediate the hormonal responses that orchestrate the hunger–satiety cycle [44].

Enteroendocrine cells comprise 90% of all intestinal epithelial cells and are polarized such that they permit the transport of nutrients from the gut lumen through apical sodium-glucose cotransporter-1 (SGLT-1) and into circulation through glucose transporter-2 (GLUT2) [45]. The hormones secreted by EECs such as GLP-1, PYY, and CCK can act locally as paracrine factors, neurotransmitters, and neuromodulators, or enter the bloodstream and act as classical hormones at distant sites [46,47]. It has been established that SGLT-1 based transport is critical for GLP-1 release in humans which enhances glucose-induced insulin secretion from pancreatic β-cells [45,48]. In animals, several sweet stimuli including NNS have been shown to upregulate SGLT-1 expression and function, suggesting that SGLT-1 activity is modulated by an upstream and broad sweet taste receptor [49,50]. Thus, it is thought that NNS can potentiate SGLT-1 function and glucose absorption [51]. NNS including sucralose and Ace-K demonstrate high levels of GLP-1 secretion in in vitro studies, with many inconclusive results in human studies [52]. Given the collective effects of these hormones, it is likely that they contribute to the pathogenesis of metabolic disorders, including obesity and T2DM [19,47,53]. Thus, it is possible that NNS can stimulate sweet-taste receptors on intestinal EECs to promote the release of these hormones involved in glucose homeostasis [26,48].

## 6. Non-Nutritive Sweeteners Interfere with Gut Microbiota Composition

The gut microbiota consists of millions of bacteria, viruses, and fungi that exist symbiotically within the gut and begins to develop at birth [54]. The composition and function of the microbiota varies not only amongst individuals, but also changes throughout an individual’s life, affected by external factors such as environmental stressors, antibiotics and diet [55]. It is thought that diet is responsible for approximately 10% of the influence on intestinal microbiota, a substantial amount when considering the high variability in lifestyle and genetics amongst individuals [56]. Aberrations in the gut microbiota have been associated with the development of insulin resistance, obesity, and metabolic syndrome; however, the details are still in the process of being understood [46,57]. In particular, it has been reported that T2DM is associated with alterations in microbiota composition [58].

In the human gut, the most common phyla are the Gram-positive *Firmicutes* and the Gram-negative *Bacteroidetes* [59]. Analysis of the gut microbiota in lean and obese individuals has revealed differences in the phyla present. There are several reports on a higher ratio of *Firmicutes* to *Bacteroidetes* in obese individuals compared to lean individuals, with the proportion of *Bacteroidetes* increasing with weight loss [60,61,62]. As a result, it has been speculated that the differences in the phyla present may be associated with the development of obesity, a component of metabolic syndrome. However, there are conflicting results, and specific roles of phyla have not yet been fully established [60]. Given the differences in microbiota composition amongst lean and obese individuals and the negligible caloric value of NNS, it is surprising that NNS consumption may induce changes in microbiota composition [52]. There have been several forms of dysbiosis that have been observed following NNS consumption, mainly an increased ratio of *Firmicutes*:*Bacteroidetes* and an increase in *Lactobacilli* spp., such that the microbiota composition resembles that of obese individuals [63].

Suez and colleagues first reported the dysbiosis that occurs as a result of NNS consumption in animal studies [63]. There are several diet-induced animal models of metabolic syndrome, in which the animals are fed a single type or a combination of diets, investigating the whole-body effects of metabolic syndrome such as through hormones, glucose metabolism and lipid metabolism pathways [64]. Suez et al. reported on the cooperation between microbial species in the gut being linked to enhanced energy harvest that promotes lipogenesis in mice through glycan degradation pathways [63]. Interestingly, the metagenomes of saccharin-consuming mice were found to be enriched with pathways such as sphingolipid metabolism and lipopolysaccharide biosynthesis, both of which have been associated with T2DM and obesity [65,66]. Perhaps the most intriguing result of the study was that the *Bacteroidetes* to *Firmicutes* ratio was positively correlated with reduced glucose tolerance, and the reverse tendency was observed for overweight people, and the deleterious metabolic effects were transferable to germ-free mice [67]. Thus, it is essential that we consider the gut microbiota composition when developing treatment strategies for T2DM and obesity within the metabolic syndrome platform.

NAS-induced gut microbiota composition changes have been linked to the phenomenon of metabolic endotoxemia, the development of a low-grade inflammatory state by the gut microbiota that ultimately promotes the development of insulin resistance (Figure 2) [68]. Briefly, dead bacteria result in the release lipopolysaccharides (LPS) into the gut. LPS is absorbed into circulation where it binds to CD14 proteins (modulators of insulin sensitivity in animals with hyperglycemia, hyperinsulinemia, and weight gain), nucleotide oligomerization domains (NODs), and Toll-like receptors (TLRs) on the surface of the macrophages and dendritic cells. The activation of these innate immune cells initiates several inflammatory processes through the release of inflammatory cytokines [69]. Overproduction of inflammatory cytokines, in turn, activates additional signaling pathways in metabolic cells that ultimately result in insulin desensitization, altered expression of proteins responsible for glucose transport, increased intestinal permeability, LPS infiltration, oxidative stress, and adipose tissue inflammation [68]. Metabolic endotoxemia may be a driving force behind NAS-induced obesity and insulin resistance.

### 6.1. The Role of GPCR-IR Crosstalk in Metabolic Syndrome

There has been extensive research elucidating the role of diet in modulating gut microbiota composition in humans and animals [55,70,71]. Specifically, it has been suggested that NNS consumption modulates gut microbiota composition as it has been associated with an increased risk of obesity, T2DM, and metabolic syndrome [7,72]. As a result, there is increasing interest to understand the signaling pathways implicated in metabolic syndrome, with a particular focus on the novel phenomenon of biased agonism of GPCRs. Here, specific substrates or metabolites can induce the preferential activation of these specific GPCRs [68,73]. Indeed, the gut microbiota can produce short-chain fatty acids (SCFAs) as metabolites from the host diet that bind to specific GPCRs [74,75,76] and confer insulin resistance through biased activation of insulin receptor (IR) signaling [77]. Also, diet- and NNS-induced gut microbiota composition changes have been linked to metabolic syndrome. This process may be mediated at least in part by miRNAs that can enter bacterial mitochondria, regulating their gene expression and ultimately promoting a state of dysbiosis that can result in the development of insulin resistance and T2DM in the host [78].

As discussed, the development of insulin resistance, obesity, and overall metabolic syndrome has been associated with changes in gut microbiota composition. The sweet taste receptors are associated with G protein α-gustducin, and GPCR crosstalk with several receptors, and particularly, the insulin receptor (IR) has been implicated in the development of metabolic syndrome [77]. The receptors of the taste buds are coupled to G proteins (T1R2 and T1R3), forming part of the C class of GPCRs, which are structurally similar to the glutamate metabotropic receptors.

GPCR-IR crosstalk has been associated with altering gut motility and permeability of SCFAs [79]. Briefly, SCFA bind to GPCRs and ultimately get into systemic circulation and activate several metabolic and inflammatory processes [79]. The crucial role of GPCR proteins from studies with Gpr41-deficient mice which were significantly leaner than control mice have validated the role of GPCR proteins [73]. Gpr41 and Gpr43 are a pair of mammalian orphan GPCRs expressed in human adipocytes, colon epithelial cells, and peripheral blood mononuclear cells. These GPCRs are activated by SCFAs such as acetate, propionate, and butyrate, which are produced during dietary fiber fermentation by resident gut bacteria. SCFA stimulation of Gpr41 increases leptin release, which slows gut motility and promotes lipogenesis [80]. Gpr43, another SCFA receptor, was found to inhibit insulin signaling in adipose tissue, suppressing fat accumulation and promoting weight loss in mice [81]. Indeed, GPCRs are closely linked to microbe–microbe and microbe–host interactions that play a role in processes such as energy harvest, storage, and expenditure. Since NAS play a role in changing the gut microbiota composition, it is possible that GPCR crosstalk with other receptors is a mechanism by which NAS consumption contributes to the development of metabolic and immunological abnormalities.

We have previously reviewed the novel phenomenon of GPCR-biased agonism or functional selectivity and its role in the development of metabolic syndrome [77]. In particular, Haxho et al. discovered that specific ligands signaling through the neuromedin B receptor result in preferential IR activation as a result of GPCR-biased agonism [82]. They found that the GPCR agonists, angiotensin and bradykinin dose-dependently induced neuraminidase-1 (Neu-1) sialidase activity through matrix metalloproteinase-9 (MMP9) activation. Activated Neu-1 allows for insulin receptor dimerization, leading to intracellular insulin signaling cascade. This concept explains how GPCR-MMP9-IR crosstalk contributes to the development of insulin resistance through the over-activation of insulin receptor signaling without its ligand [82] (Figure 3). It can also potentially explain the mechanism of NAS-induced metabolic changes, and how they may lead to the development of T2DM via the novel phenomenon of functional selectivity [83].

An intriguing observation about the T1R2/T1R3 heterodimeric receptor is its unique structural diversity for a vast assortment of ligands. As discussed, the sweet taste receptor is able to recognize every sweetener available, including carbohydrates, amino acids and their derivatives, proteins, and synthetic sweeteners [84]. Interestingly, this receptor can exhibit unique stereoselectivity for certain molecules such as D-tryptophan but not L-tryptophan [84]. This stereoselective property of the T1R2/T1R3 receptor, where it can adopt more than one active state, is the phenomenon of ‘functional selectivity’, ‘ligand directed signaling’, or ‘biased agonism’ [85,86,87,88,89,90]. There are also allosteric ligands with different degrees of modulation, called ‘biased modulation’, that can dramatically influence GPCRs in a probe- and pathway-specific manner [85,88,89], including the T1R2/T1R3 receptor. For example, the human and rodent sweet taste receptors exhibit differences in ligand specificity, G protein-coupling efficiency, and sensitivity to inhibitors [30]. Li et al. [84] provided evidence that both the human and rat taste receptors can couple efficiently to Gα_15/i1_, but only the human receptor can couple efficiently to Gα_15_. Xu et al. [30] also provided evidence to support the important role of T1R2 in Gα-protein coupling in a functional expression system. To explain these observations, Onfroy et al. proposed that G protein stoichiometry dictates biased agonism through distinct receptor–G-protein partitioning [91]. Here, expression levels of Gα subunits influence the biased profiling of agonists as well as antagonists, such that they determine both their activity and G protein coupling efficacy by affecting different membrane distribution of receptor–G protein populations. In the naïve state, the level of Gα expression influences the partitioning of not only Gα but also the co-expressed receptor in different membrane domains [91]. It is intriguing to speculate that the T1R2/T1R3 taste receptor through Gα protein partitioning involves a “pluridimensional efficacy” concept as previously described by Galandrin and Bouvier [92] for distinct signaling profiles of β1/β2 adrenergic GPCR receptor ligands.

Here, the potential mechanism(s) that might connect NNS to metabolic changes is the molecular crosstalk between the IR and GPCRs through a putative pluridimensional Gα protein partitioning efficacy. Hypothetically, the heterodimeric T1R2/T1R3 GPCR taste receptors could exist in a multimeric receptor complex with NMBR, IRβ, and Neu1 in naïve IR expressing cells as depicted in Figure 3. In this situation, the TIR2/TIR3 dimeric receptors would exist as a molecular link regulating the interaction and signaling mechanism(s) between these molecules on the cell surface. This hypothetical molecular model could explain a biased TIR2/TIR3 GPCR agonist-induced IRβ transactivation signaling axis, mediated by Neu-1 sialidase activity and the modification of insulin receptor glycosylation.

Collectively, these studies provide strong evidence that there exists an interaction between GPCRs and the insulin receptor, which may provide insight into the implications of NNS consumption on the development of metabolic syndrome. In summary, consumption of NNS may activate GPCR signaling pathways that lead to cross-activation of insulin receptors through the neuromedin B receptor (NMBR) and ultimately promote the development of insulin resistance and T2DM.

### 6.2. NAS Modify miRNAs in Regulating Gut Composition

Recent attention on the role of miRNAs changing cell function has incited interest in miRNA implications on gut microbiota function, and consequently the development of insulin resistance in individuals consuming NNS in their diet. miRNAs are 18–25 nucleotide long noncoding RNAs that alter gene expression through post-transcriptional silencing or activation [93]. It is speculated that miRNAs regulate at least 30% of human genes, playing critical roles in cell proliferation, differentiation, apoptosis, and hematopoiesis [94,95]. As a component of human feces, miRNAs can be biomarkers, prognostic indicators, and regulators of normal and abnormal cell function [78,96]. Preclinical models suggest that miRNA expression can be altered by stress, exercise and diet [97]. As a result, NNS consumption may modify miRNA expression by altering bacterial composition and potentially lead to metabolic changes.

Liu et al. recently elucidated the mechanism by which miRNAs shape gut microbiota composition [78]. Intestinal epithelial cells (IECs) secrete miRNAs into the gut lumen in the form of exosomes or extracellular vesicles. The secreted miRNAs can then enter gut bacteria and act at the DNA level or directly on RNA in the mitochondria where they can alter gene expression that regulates functions affecting bacterial growth. Fecal transplantation of miRNA has been reported to help restore gut microbiota composition, broadening the therapeutic application of miRNA in influencing the abundance of health-associated gut bacteria.

The role of miRNAs in the pathogenesis of glucose intolerance and T2DM has been increasingly investigated and explained in detail elsewhere [98]. Briefly, miRNAs were shown to play critical roles in core processes of the insulin-related signaling pathway, carbohydrate and lipid metabolism, as well as adipocytokine signaling pathways. Up- or downregulation of certain miRNAs has been correlated with the development of insulin resistance and increased severity of T2DM. In particular, miR-7 was found to regulate pancreatic β-cell function, differentiation and insulin secretion [99]. Overexpression of miR-7 in mice has been associated with the development of T2DM due to impaired insulin secretion. Also, levels of miR-101, miR-375, miR-802 were significantly increased in T2DM patients while miR-143 and miR-223 levels were downregulated in obese individuals when compared to control groups [100,101]. Constant exposure of pancreatic β-cells to various metabolic stresses, including NNS consumption, can shift the delicate balance between positive and negative regulatory miRNA, ultimately promoting pancreatic dysfunction and insulin resistance. Importantly, miRNAs can affect different bacterial species due to their capacity to enter the GI tract [102]. This effect can partially explain the differences observed in miRNA regulatory effects. If there is a prevalence of particular phyla in the gut, they can modify the expression of genes that correspond to changes in metabolic functions, including endocrine cell dysfunction. Several studies have demonstrated that regular NNS consumption induces changes in the composition of the gut microbiota that eventually leads to the development of insulin resistance [103,104]. The driving force behind this pathological process may be miRNA regulation of insulin-related signaling pathways.

Circulating miRNAs can be used as diagnostic biomarkers for T2DM patients; however, there are some challenges in assessing their levels [105]. Current biomarkers only detect disease once metabolic imbalances have already set in, whereas changes miRNA expression levels can be noticed 5–10 years before T2DM manifestation [106]. The small size and known physiochemical properties of miRNAs alongside their natural synthesis make them attractive therapeutic targets [96]. Current approaches for modulating miRNA function *in vivo* demonstrate promising results. As regulators of many metabolic processes, miRNAs have the potential to be therapeutic agents [107]. For example, miR-126 expression is significantly reduced in diabetic patients, leading to impaired proangiogenic capacity that promotes diabetic vasculopathy. Some experiments have been able to manipulate miR-126 expression to induce migration and proliferation of vascular endothelial cells and facilitate their repair [108]. Collectively, it is imperative that technology continues to advance in this field to better assess the effects of miRNA-based therapies.

## 7. Non-Nutritive Sweeteners Interfere with Learned Responses to Sweetness

Sugar and its sweet-tasting nutritive and non-nutritive alternatives have become a staple in the diet. However, sweet-taste has been associated with learned behavior [109]. As discussed previously, sugar consumption has been associated with an increased GLUT2 and GLUT5 expression, which play a role in CCK expression in the ileum of isocaloric diet-fed rats enriched with fructose or glucose [110]. The enriched diets provide additional calories, resulting in animals having enhanced total caloric intake [111]. In contrast to natural sweeteners such as fructose or sucrose, NNS was thought to be excreted after passing through the GI tract unchanged resulting in no energy gain [112].

Theoretically, the metabolic effects observed with the use of natural sweeteners should be absent with NNS consumption. Paradoxically, NNS consumption has been associated with weight gain. It is hypothesized that the separation of sweetness from calories interferes with physiological responses and the interaction of NNS with sweet-taste receptors in the gut that affect glucose absorptive capacity and homeostasis [113,114]. Although epidemiological studies have shown an association between artificial sweetener use and weight gain, evidence of a causal relationship is limited; however, recent animal studies provide intriguing information that supports an active metabolic role of artificial sweeteners [19]. Indeed, the low or zero caloric value of NNS can result in caloric compensation, whereby there is an adjustment for calories consumed at one occasion by reducing caloric intake at subsequent opportunities. Thus, weakened caloric compensation can result in excess energy intake that ultimately leads to increased weight gain [115].

In 1910, Pavlov and more recently by other studies [116] proposed that orosensory stimuli like sweet taste elicit different learned physiological responses, including the anticipation of the arrival and absorption of food to control body weight and energy balance. Interestingly, there might be a role for gustatory cues in the detection of high fat/high sugar diets [116]. The fat component is a more salient orosensory feature of the high energy diet. High caloric compensation is observed when rats are fed glucose, but there is weaker compensation when rats are fed saccharin [117]. If these animal studies had a longer duration, the reduced caloric compensation would result in increased weight gain. Non-caloric sweeteners reduce the validity of sweet taste as a signal to predict caloric intake that leads to a positive balance of energy and weight gain. Green et al. conducted imaging studies using fMRIs and found alterations in the reward processing of sweet taste in the individuals who regularly consumed diet soda [118]. More human studies are required to assess physiological responses to NNS in naïve subjects.

Weight gain associated with NNS consumption can be explained in part by their interference with learned responses that contribute to energy homeostasis. Swithers and Davidson demonstrated that NNS consumption weakens cephalic response to ingested food [119]. Based on the Pavlovian conditioning principles, it is hypothesized that sweet taste predicts energy intake and evokes both autonomic and endocrine responses that prepare the GI tract [120]. Altered glucoregulatory responses, such as the release of GLP-1, were observed in mice when glucose and saccharin were given orally but not when directly released into the stomach [120]. These findings support the proposition that there is a disruption in the learned responses generated by tasting sweetness but not in the post-absorptive consequences of consuming sugar. Suppressed GLP-1 release when saccharin is given orally may disrupt the satiety process due to increased gastric emptying. Lower GLP-1 levels lead to decreased glucose utilization in muscle, liver, adipose tissue, and diminished suppression of glucagon release, elevating blood glucose levels. Considering that cephalic responses are required for a normal postprandial glucose tolerance [121], it is possible that NNS interferes with learned responses to sweetness, and thereby increasing food intake and weight gain.

In addition, M.O. Welcome reviewed recent evidence to demonstrate that the sweet taste receptor heterodimer T1R2/T1R3 plays a crucial role in cognitive functioning, suggesting that dysfunctions in sweet taste receptor signaling may underlie cognitive impairment in some brain pathologies [122]. Dysfunctions in sweet taste receptor signaling were also associated with inflammatory response pathway [123]. The study showed that sweet taste receptors function as pivotal immune sentinels, revealing the downregulation of the key components of the taste signaling cascades such as α-gustducin, phospholipase C β2, and monovalent selective cation channel TRPM5, contributing to cognitive impairment [123].

## 8. Conclusions

Non-nutritive sweeteners continue to be a staple in the Western diet. However, the health-related safety of NNS consumption remains to be a controversial topic. Recent reports on the role of NNS promoting shifts in gut microbiota composition and reports linking the gut microbiota to insulin signaling, confirm the importance of studying the physiological effects of NNS. The exact mechanisms of pathological changes induced by NAS are still under speculation. Thus, the recently discovered phenomenon of functional selectivity of GPCR-IR crosstalk, as well as miRNA modulation of gut microbiota function, may provide insight into the pathological effects of NNS. The proposed mechanisms underlying NNS-mediated development of metabolic syndrome are summarized in Figure 1. Enhancing our knowledge through well-designed human trials should highlight the potential role of NNS in the alterations of microbial, neurological, and hormonal responses to consumed food. Energy intake compensation appears to be an area where additional studies need to test and compare different food as well as different NAS. Consumers must be aware that contrary to the existing belief, that substitution of natural sugar by NAS is beneficial for their health, there is growing evidence of NAS being implicated in the development of metabolic abnormalities. Continued research in this field will uncover the pathology of diet-induced metabolic changes as well as uncover new biomarkers and novel treatments using miRNAs.

## Figures and Tables

**Figure 1 nutrients-11-00644-f001:**
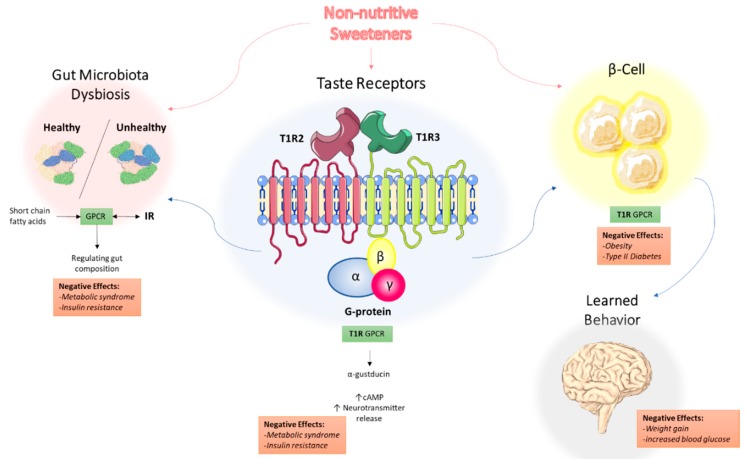
Proposed mechanisms of the underlying effects of non-nutritive sweeteners on the development of metabolic syndrome. NNS interact with the T1R family of sweet-taste receptors through associated G protein α-gustducin, which results in increased intracellular cAMP levels and increased neurotransmitter release. Through the associated GPCR signaling, this may explain how NNS can contribute to metabolic syndrome and insulin resistance. NNS also interfere with gut microbiota composition, with short-chain fatty acids (SCFAs) from dietary intake acting as ligands for GPCRs in the gastrointestinal tract, regulating NNS permeability and gut microbiota composition. Additionally, NNS are associated with insulin and other hormone secretion, which ultimately impact learned behavior and response to sweetness. Abbreviations: NNS, non-nutritive sweeteners; GPCR, G protein-coupled receptor; SCFA, short-chain fatty acid.

**Figure 2 nutrients-11-00644-f002:**
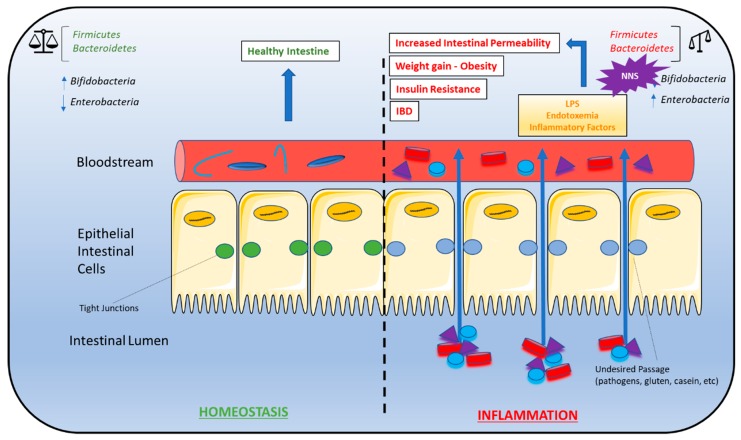
Gut microbiota dysbiosis and metabolic syndrome. Dysbiosis of the *Firmicutes:Bacteroidetes* ratio is associated with several conditions characteristic of metabolic syndrome, including weight gain/obesity, insulin resistance, high-fat diets, gut permeability, and inflammatory bowel disease (IBD). As a result, NNS consumption may contribute to the development of these conditions due to alterations in the *Firmicutes:Bacteroidetes* ratio. A *bifidobacteria* decrease combined with an *enterobacteria* increase leads to endotoxemia that causes a chronic low-grade inflammation associated with some pathological conditions such as insulin resistance and increased gut permeability. A right balance in the microbiota may be considered in gut homeostasis and maintaining the microbiota can be considered prebiotics and restore eubiosis in some pathological conditions. Abbreviations: IBD, inflammatory bowel disease; NNS, non-nutritive sweeteners.

**Figure 3 nutrients-11-00644-f003:**
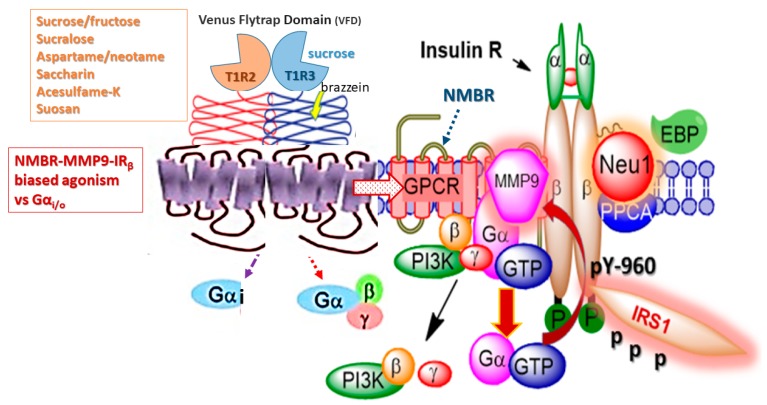
Hypothetical G protein-coupled receptor (GPCR) heterodimeric T1R2/T1R3 taste receptors exist in a multimeric receptor complex with NMBR, IRβ, and Neu-1 in naïve IR expressing cells. Here, TIR2/TIR3 dimeric receptors are proposed to exist as a molecular link regulating the interaction and signaling mechanism(s) between these molecules on the cell surface. This molecular model uncovers a biased TIR2/TIR3 GPCR agonist-induced IRβ transactivation signaling axis, mediated by Neu-1 sialidase and modification of insulin receptor glycosylation. This novel biased GPCR-signaling platform potentiates neuraminidase-1 (Neu-1) and matrix metalloproteinase-9 (MMP-9) crosstalk on the cell surface that is essential for the activation of the insulin receptor β subunit (IRβ) tyrosine kinases. Notes: Insulin-binding receptor α subunits (IRα), as well as GPCR agonists, potentiate biased neuromedin B receptor (NMBR)-IRβ signaling and MMP-9 activation to induce Neu-1 sialidase. Activated MMP-9 is proposed here to remove the elastin-binding protein (EBP) as part of the molecular multienzymatic complex that contains β-galactosidase/Neu-1 and protective protein cathepsin A (PPCA). Activated Neu-1 hydrolyzes α-2,3 sialyl residues of IRβ at the ectodomain to remove steric hindrance to facilitate IRβ subunits association and tyrosine kinase activation. Activated phospho-IRβ subunits phosphorylate insulin receptor substrate-1 (pIRS1), which initiate intracellular insulin signaling via the Ras-MAPK and the PI3K-Akt pathway, among others. Abbreviations: PI3K, phosphatidylinositol 3-kinase; GTP, guanine triphosphate; IRS1, insulin receptor substrate-1; p, phosphorylation; Neu-1, neuraminidase-1. Taken in part from Cellular Signalling 43 (2018) 71–84. © 2018 Haxho et al., Published by Elsevier Inc., open-access under CC BY-NC-ND license. This is an open-access article which permits unrestricted noncommercial use, provided the original work is properly cited.

**Table 1 nutrients-11-00644-t001:** Classification of Food and Drug Administration (FDA)-approved sweeteners.

Name	Brand Names	Applications in Food Industry	Relative Sweetness (Measured to Sucrose)
**High-intensity Sweeteners**			
**Saccharin**	Sweet and Low^®^, Sweet Twin^®^, Sweet’N Low^®^, Necta Sweet^®^	Beverages, bases, and mixes for many food products, table sugar substitute	200–700×
**Aspartame ***	Nutrasweet^®^, Equal^®^, Sugar Twin^®^	Soft drinks, chewing gum, pudding, cereals, instant coffee Also distributed as a “General Purpose Sweetener”	200×
**Acesulfame-potassium (Ace-K)**	Sunett^®^, Sweet One^®^	Beverages, candy, frozen desserts, baked goods Heat stable so it can be used in baking	200×
**Sucralose**	Splenda^®^		600×
**Neotame**	Newtame^®^	Beverages, candy gum	7000–13,000×
**Advantame**	N/A	Baked goods, beverages, frozen desserts, frosting, chewing gum, candy, pudding, jelly and jam, gelatin	20,000×
**Sugar Alcohols**			
**Erythritol**		Fondant, ice cream, gum, tabletop sweeteners, chocolate, dairy products, jelly, beverages	0.60×–0.70×
**Glycerol**		Dairy products, processed fruits, energy bars, jam, fondant Often used as a thickening agent and to provide texture to food	
**Mannitol**		Infant formula, frozen fish, precooked pasta, butter, chocolate flavored coatings	0.50×–0.70×
**Sorbitol (Glucitol) ***		Used as emulsifier	0.66×
**Xylitol**		Hard candy, chewing gum, mints, ice cream, chocolate, cookies, beverages, table sugar substitute	1×
**Natural Sweeteners**			
**Steviol glycosides**	Natural constituents of leaves of *Stevia rebaudiana* (Bertoni) plant, commonly known as Stevia	Beverages, chewing gum, candy	200–400×
**Luo Han Guo Monk fruit extracts**	*Siraitia grosvenorii* Swingle fruit extract (SGFE)	Tea	100–250×
**Lucuma powder**		Beverages, pudding, granola, pastry, baked goods	

* Nutritive sweetener. Content taken in part from the FDA approval of artificial sweeteners. https://www.fda.gov/food/ingredientspackaginglabeling/foodadditivesingredients/ucm397725.htm and Shwide-Slavin et al. [11].

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
