# Peer review of "Non-Nutritive Sweeteners and Their Implications on the Development of Metabolic Syndrome"

_nutrients, 2019, doi:10.3390/nu11030644_

Round 1

Reviewer 1 Report

This is an interesting manuscript which evaluates the potential impact of NNS on the development of the metabolic syndrome. Replacing sugar-sweetened products with products containing sugar substitutes (NNS) that provide few or no calories has been used as a strategy for supporting public health outcomes. However, current scientific evidence indicates that frequent intake of foods with NNS not only fails to prevent diet-related chronic diseases, but is associated with increased risk of some health outcomes. These include gut microbiota dysbiosis and increased likelihood of glucose intolerance which is associated with increased risk of developing type 2 diabetes. Therefore, convincing data in this area could have an impact on nutritional practice.

This is an interesting review with a reasonable interpretation of results. While there are a number of limitations that affect the assessment of biologic causality, the presented data is very interesting. A few comments and suggestions are listed below.

Introduction:  The introduction is too long and multithreaded. It should be shortened and a part of it should be extracted into a new section, e.g. Current Use of Non-Nutritive Sweeteners.  In the new section - Current Use of Non-Nutritive Sweeteners, reference should be made to Table 1 which should be corrected according to the suggestions below.

Table 1 (line 85): Sweeteners presented in the Table should be divided into Synthetic high-intensity sweeteners and Natural high-intensity sweeteners. Column “Type” should be deleted as all sweeteners are taken as NNS. For steviol glycosides and Luo Han Guo, the examples of brand names should be given. Please replace “Comparison to sucrose” with - Relative sweetness (measured to sucrose). Additionally, please add a new column to the table – Applications in food industry, and fill it in with relevant examples of products.   

Figure 1 (line 231) – please correct and ensure that NNS are included.

In order to visualize better the three potential mechanisms and the specific role of NNS (NNS interacting with sweet taste receptors; NNS interfering with gut microbiota composition, and NNS interfering with learned responses to sweetness), I suggest to present them in a new figure, e.g. Proposed mechanisms involving NNS underlying the development of metabolic syndrome. This would significantly improve the value of the whole manuscript.

Author Response

"Introduction:  The introduction is too long and multithreaded. It should be shortened and a part of it should be extracted into a new section, e.g. Current Use of Non-Nutritive Sweeteners.  In the new section - Current Use of Non-Nutritive Sweeteners, reference should be made to Table 1 which should be corrected according to the suggestions below."  Author response: Thank you for this comment. We have shorten the introduction and included the suggested new section and Table 1.

"Table 1 (line 85): Sweeteners presented in the Table should be divided into Synthetic high-intensity sweeteners and Natural high-intensity sweeteners. Column “Type” should be deleted as all sweeteners are taken as NNS. For steviol glycosides and Luo Han Guo, the examples of brand names should be given. Please replace “Comparison to sucrose” with - Relative sweetness (measured to sucrose). Additionally, please add a new column to the table – Applications in food industry, and fill it in with relevant examples of products." Author response: Thank you for this comment. We have revised Table 1 accordingly.

"Figure 1 (line 231) – please correct and ensure that NNS are included. " Author response: Thank you for this comment. Figure 1 in original draft is now Figure 2. We have included NNS in the figure as recommended.

"In order to visualize better the three potential mechanisms and the specific role of NNS (NNS interacting with sweet taste receptors; NNS interfering with gut microbiota composition, and NNS interfering with learned responses to sweetness), I suggest to present them in a new figure, e.g. Proposed mechanisms involving NNS underlying the development of metabolic syndrome. This would significantly improve the value of the whole manuscript. " Author response: Thank you for this excellent comment. We have included Figure 1 to visualize better the three potential mechanisms. Also, we have included another Figure 3 visualizing the hypothetical G-protein-coupled receptor (GPCR) heterodimeric T1R2/T1R3 taste receptors exist in a multimeric receptor complex with NMBR, IRβ, and Neu-1 in naïve IR expressing cells. This part of the text  was deemed weak in the original draft, and we carefully articulated how T1R2.T1R3 taste receptors might exist in a multimeric complex of IR contributing to metabolic syndrome and IR resistance. It is important to add this to the manuscript.

Reviewer 2 Report

The authors have written an interesting review on the impact NAS/NNS on the metabolic syndrome, and lists references describing mechanisms for how NAS/NNS can affect physiology. The review is well written and includes key references. One suggestion would be to include one more figure at the end, which summarizes and emphasizes the main conclusions of the review (involved mechanisms for NAS/NNS on health). 

Author Response

The authors have written an interesting review on the impact NAS/NNS on the metabolic syndrome, and lists references describing mechanisms for how NAS/NNS can affect physiology. The review is well written and includes key references. One suggestion would be to include one more figure at the end, which summarizes and emphasizes the main conclusions of the review (involved mechanisms for NAS/NNS on health).

Authors response: Done. Excellent suggestion and it is now Figure 1.